# Nematode Community-Based Soil Food Web Analysis of Ferralsol, Lithosol and Nitosol Soil Groups in Ghana, Kenya and Malawi Reveals Distinct Soil Health Degradations

Haddish Melakeberhan [1,*], ZinThuZar Maung [1,2], Isaac Lartey [1], Senol Yildiz [1,3], Jenni Gronseth [4], Jiaguo Qi [4], George N. Karuku [5], John W. Kimenju [6], Charles Kwoseh [7] and Thomas Adjei-Gyapong [7]

1 Agricultural Nematology Laboratory, Department of Horticulture, Michigan State University (MSU), East Lansing, MI 48824, USA; zmaung@ucr.edu (Z.M.); larteyis@msu.edu (I.L.); s7yildiz@gmail.com (S.Y.)
2 Nematology Laboratory, Kearney Agricultural Research and Extension Center, Department of Nematology, University of California (Riverside), Parlier, CA 93648, USA
3 Department of Plant Protection, K.T. Manas University, Bishkek 72004, Kyrgyzstan
4 Center for Global Change and Earth Observations, MSU, East Lansing, MI 48824, USA; jenni.gronseth@gmail.com (J.G.); qi@msu.edu (J.Q.)
5 Department of Land Resource Management and Agricultural Technology, University of Nairobi, Nairobi P.O Box 30197, Kenya; g_karuku@yahoo.com
6 Department of Plant Science and Crop Protection, University of Nairobi, Nairobi P.O Box 30197, Kenya; wkimenju@yahoo.com
7 Department of Crop and Soil Science, Kwame Nkrumah University of Science and Technology, PMB, UPO, Kumasi, Ghana; ckwoseh@hotmail.com (C.K.); djgypng@gmail.com (T.A.-G.)
* Correspondence: melakebe@msu.edu

**Abstract:** Determining if the vast soil health degradations across the seven major soil groups (orders) of Sub-Saharan Africa (SSA) can be managed on the basis of a one-size-fits-all or location-specific approach is limited by a lack of soil group-based understanding of soil health degradations. We used the relationship between changes in nematode population dynamics relative to food and reproduction (enrichment, EI) and resistance to disturbance (structure, SI) indices to characterize the soil food web (SFW) and soil health conditions of Ferralsol, Lithosol and Nitosol soil groups in Ghana, Kenya and Malawi. We applied bivariate correlations of EI, SI, soil pH, soil organic carbon (SOC), and texture (sand, silt and clay) to identify integrated indicator parameters, and principal component analysis (PCA) to determine how all measured parameters, soil groups, and countries align. A total of 512 georeferenced soil samples from disturbed (agricultural) and undisturbed (natural vegetation) landscapes were analyzed. Nematode trophic group abundance was low and varied by soil group, landscape and country. The resource-limited and degraded SFW conditions separated by soil groups and by country. EI and SI correlation with SOC varied by landscape, soil group or country. PCA alignment showed separation of soil groups within and across countries. The study developed the first biophysicochemical proof-of-concept that the soil groups need to be treated separately when formulating scalable soil health management strategies in SSA.

**Keywords:** biodiversity; degradation; ecosystems; nematodes; small-holder agriculture; soil health; Sub-Saharan Africa

## 1. Introduction

The impact of the vast degradations of soil health further confounded by climate change on food security and livelihood of the increasing population in Sub-Saharan Africa (SSA) is well documented [1–4]. Soil health, defined as a given soil's ability to generate desirable ecosystem services, comprises biological, nutritional, physicochemical, structural, and water holding capacity components that need to be kept in balance at all times [5,6]. Despite a substantial knowledge on the soil health components and agroecosystem function

and services [7–12], developing soil health management models that fit the vast soil degradations in SSA is limited by a) variable standards and outcomes, b) existing knowledge lacking an integrated platform, and c) issues of scale within and across regions where the levels of soil degradations, cropping systems, land use practices, and cultures are highly variable [4,12]. Therefore, developing soil health management strategies to that fit the extent of SSA soil degradations will require identifying if the degradations across space fit a one-size-fits-all or a location-specific approach to fix. This paper characterizes the types and levels of soil health degradations in soil groups (orders) and provides a proof-of-concept for a location-specific approach to formulating soil health management strategies in SSA.

Recent studies and meta-analyses of the relationships between soil health and ecosystem services in SSA and elsewhere showed highly variable standards and outcomes [10–16]. For example, soil health indicators in a given soil may be biological, chemical and other parameters [11–13], but the spatial heterogeneity of soils, land use practices and cropping systems, make it difficult to standardize the biophysicochemical outcomes across regions and sociocultures [12,14]. As used here, biophysicochemical changes refer to a combination of biological, chemical, physical, and nutritional components of soil health.

An integrated understanding of the biophysicochemical changes in soil degradation remains challenging because there is no standard platform where the different components of soil health can be assessed. In this regard, the soil food web (SFW), in which nematodes are key players, is central to understanding biophysicochemical changes including nutrient cycling and soil health in response to land use and agronomic practices [5,17,18]. As the most abundant metazoan on the planet, nematodes are key indicators of changes in soil and belowground ecosystems at two levels [5,19–25]. First, when nematodes are identified to herbivore (HV), bacterivore (BV), fungivore (FV), predator (PR), and omnivore (OV) trophic groups and colonizer-persister (c-p) as c-p 1 (short life cycle and less sensitive to disturbance) to c-p 5 (long life cycle and sensitive to disturbance) life strategies [26,27], they can reveal information on community abundance and diversity, ecosystem disturbance, and the quality of the environment [28–31]. Second, changes in nematode population densities can reveal the structure and function of the SFW and nutrient cycling because nematodes feed on microorganisms and they are food for other organisms [20–31]. Nematode population densities change as a function of food resource availability and reproduction strategies as well as resistance to disturbance of their environment. To visually describe the SFW conditions, Ferris et al. [28] plotted Enrichment Index (EI; *y*-axis), which measures changes in population density of nematodes by functional guilds relative to their food resource and reproductive rate, against the Structure Index (SI; *x*-axis), which measures changes in these nematode functional guild population densities relative to their resistance to disturbance. In this way, four SFW conditions are identified in x-y quadrants: enriched but unstructured (Quadrant A; negative-x, positive-y), enriched and structured (Quadrant B; positive-x, positive-y), resource-limited and structured (Quadrant C, negative-x, positive-y), or resource-depleted with minimal structure (Quadrant D; negative-x, negative-y). Enriched means N is available and depleted means N is held in the organisms. Quadrant B is desirable for agroecosystems where the SFW condition is nutrient enriched and balanced in the decomposition process [28]. Even if the Ferris et al. [28] SFW model can be a tool for identifying the soil conditions through an integrated understanding of the belowground biophysicochemical changes and making decisions on developing suitable soil health management strategies, scaling up to regional, continental, and global level remains a challenge.

One way of developing scalable soil health management strategies is to use soil groups as platforms for characterizing similarities and differences in the types and levels of soil health degradations. Similar types and levels of degradation across soil groups would suggest suitability for developing a one-size-fits-all management strategy. Otherwise, location-specific approach would be the right strategy. The continent of Africa has seven major soil groups [32]. However, published information on nematodes and/or SFW where soil groups are clearly identified and inferences on soil health can be made is limited to Are-

nesols, Cambisols and Vertisols in Kenya [33,34], Quartzite in Brazil [35], and Cambisol [36], Cambisol, Chernozem, and Stognosols [25], and Cambisol Fluvisols, Rigosols, Rendizina, and Stognosols [37] in the Slovak Republic. With a long-term goal of developing soil group-based scalable soil health management strategies in SSA, this study considered the types and levels of biological degradations associated with Ferralsols, Lithosols and Nitosols in Ghana, Kenya and Malawi. The three soil groups have agroecological significance [38] that impacts over 200 million people in Africa alone. The three countries represent a strategic distribution of these soil groups that is important for building a database towards developing scalable soil health management models. The objective was to characterize the biological conditions of selected Ferralsol, Lithosol and Nitosol soil groups in Ghana, Kenya and Malawi and develop a proof-of-concept whether or not the soil conditions fit a one-size-fits-all or a location-specific approach of soil health management strategy. In developing the proof-of-concept, we tested four hypotheses that consider the limitations of variable standards, lacking an integrated platform, and scalability [7–12] in ways that will lead to integrating the different components of soil health [5,6].

Quantifying abundance of nematodes by functional guilds is a critical part of describing the SFW conditions [17]. Our first hypothesis was that the Ferralsol, Lithosol and Nitosol soil groups have different nematode community abundance and diversity. While the lack of published information on nematodes in the three soil groups makes it difficult to draw broadly applicable conclusions, the hypothesis was based on three sets of general knowledge. First, it is well established that nematode abundance and diversity vary by cropping system and climate gradient [19,20,24,31,34,35], and soil properties [25,32–34,36,37]. Second, the limited information on soil groups shows that nematode abundance and/or diversity vary by soil groups within the countries [25,32,34–37]. Third, data observed in Cambisol in the Slovak Republic [25,36,37] and Kenya [33,34] show that variability in nematode community exists across large geographic regions. If the hypothesis is true, we expect to see differences in nematode abundance among the soil groups. If false, there should be no difference in nematode abundance.

The second hypothesis was that the SFW structure of the soil groups as described by the relationship between EI and SI [28] was different across countries. The Ferris et al. SFW model [28] is based on changes in nematode population dynamics. The hypothesis is logical and consistent with nematode community abundance and composition of different structure. The intersection of the EI and SI data points describes the soil condition within a quadrant. If the hypothesis is true, the EI and SI intersection data points of the soil groups should fall in different quadrants and would indicate that soils have different SFW structures. If the hypothesis is false, the EI and SI intersection data points should fall in Quadrants A, B, C, or D and would mean the soils have similar SFW structure. Whether the hypothesis is true or false, identifying if the soil conditions are enriched but unstructured (Quadrant A), enriched and structured (Quadrant B), resource-limited and structured (Quadrant C), or resource-depleted with minimal structure (Quadrant D) will lead to developing suitable soil health management strategies through understanding of the belowground biophysicochemical changes.

There are many biophysicochemical indicators of soil health [13,16,17,21,28,29]. These include changes in micro- and macro-organism population dynamics, soil organic carbon (SOC), nutrients, pH, sand, silt, and clay texture. As pointed out earlier, correlating individual indicators does not describe the soil conditions in ways that account for all of the soil health components [5,6]. Identifying how the biophysicochemical changes may relate with EI and SI could be a step towards integrating the different soil health components. The third hypothesis was that there is no consistent correlation of EI and SI with SOC, pH, and sand, silt and clay texture. It is logical not to expect consistent correlations between EI and/or SI and the selected indicators. If the hypothesis is true, EI and/or SI will not serve as integrated indicators of the soil conditions that reflect process-based outcomes. If false, EI and/or SI along with the corresponding soil parameter will serve as integrated indicators of the soil conditions that reflect process-based outcomes.

A healthy soil is an indication that the different soil health components and below-ground processes that drive the biophysicochemical changes are in harmony [5,6,17,18]. However, there is no way of describing a healthy soil health condition from a single core of soil yet, much less scaling up the outcome across space. Knowing how the nematode abundance and diversity, EI, SI, SOC, pH, and sand, silt and clay parameters collectively align is a critical foundation for developing scalable soil health management that reflects the different soil health components. The fourth hypothesis was that there are no similarities in the alignment of the measured parameters across the three soil groups and three countries. If true, there should be no similarities in the collective alignment of all of the measured parameters across soil groups and countries. This would mean that there are location-specific conditions requiring soil group- and/or country-specific soil health degradation management strategies. If false, there should be similarities in the collective alignment of all measured parameters across soil groups and countries suitable for developing similar soil health management strategies.

## 2. Materials and Methods

### 2.1. Soil Group and Sampling Site Selection, and Design

The Ferralsol, Lithosol and Nitosol soil groups in Ghana, Kenya and Malawi were selected using a combination of FAO database [38], Google Earth, and local expert confirmation. Site selection and sampling took place during March and April 2012. In each country, two regions separated by about 15 km in Ghana and up to 300 km in Malawi were selected for each soil group. The separation of the regions was north–south direction. Hence, the regions are referred to as north and south. Soil group distribution, accessibility, and tribal and other local logistical challenges were factors in determining the distance between the regions. Within each region and soil group, one to three disturbed (agricultural and/or grazing) and an adjacent undisturbed (pristine forest and/or natural vegetation) landscapes were selected as sampling sites. In each disturbed and undisturbed site, four prelabeled flags were randomly placed and each flag was georeferenced by taking a photo with a handheld Garmin Oregon® 550 GPS unit.

### 2.2. Sampling and Sample Processing

A composite of approximately 700 cm$^3$ of soil was collected from the top 0–15 cm of three to four cores around the georeferenced flag using augers and cutlasses, where soils were too dry, and placed in a prelabeled plastic bag. A total of 152,168, and 192 georeferenced soil samples were collected in Ghana, Malawi, and Kenya, respectively. In the northern region of Ghana,4,4, and 8 from undisturbed and 8,12, and 24 samples from disturbed fields of Ferralsol, Lithosol and Nitosol, respectively, were collected. In the southern region, 12,12, and 8 from undisturbed and 16,20, and 24 samples from disturbed fields of Ferralsols, Lithosols and Nitosols, respectively, were collected. In each of the northern and southern regions of Kenya, 8 from undisturbed and samples 24 from disturbed fields of Ferralsols, Lithosols and Nitosols were collected. In the northern region of Malawi, 8, 6, and 8 from undisturbed and 22,20, and 24 samples from disturbed fields of Ferralsols, Lithosol and Nitosol, respectively, were collected. In the southern region, 8,8, and 4 from undisturbed and 24,24, and 12 samples from disturbed fields of Ferralsols, Lithosol and Nitosol, respectively, were collected. Samples were transported to the laboratory in coolers for soil physiochemistry and nematode community analyses. In the laboratory, each composite sample was thoroughly mixed to homogenize and separate soil from pieces of rocks and screened through a 4 mm sieve into a holding pan. Using a glass beaker, two separate subsamples of 100 mL of wet soil each were taken for soil and nematode community analyses [39]. The remainder of the soil was stored at 4 °C.

### 2.3. Soil Analysis

In each country, the same soil analyses procedures were followed. Percent soil organic carbon (SOC) was determined following the Destjareft method [40]. Percent sand, silt and

clay were determined following particle-size analysis [41] and soil pH was measured by the water method [42]. SOC, pH and texture were selected because they are among the closely tied soil biophysicochemical parameters to soil nutrient cycling that nematodes play a role in in the SFW.

### 2.4. Nematode Extraction and Enumeration

In each country, nematodes were extracted from 100 cm$^3$ subsamples using sugar-flotation (700 g sugar: 1 L water) and centrifugation methods for 1 min [43], fixed in double TAF solution (14 mL 40% formalin: 4 mL tri-ethanolamine: 91 mL distilled water) [44], and were shipped to Michigan State University for enumeration. Nematodes were identified to genus/family level following Bongers [45] using an inverted microscope (Motic Type 101 M, AE 2000) and assigned to either HV, BV, FV, PR, or OV trophic group [46–48] and given corresponding c-p 1 to c-p 5 values [26,27,45]. *Ditylenchus* and *Filenchus* species were classified as fungivores according to Okada and Kadota [46] and Yeates et al. [47]. Nematode data were further processed to provide information on (i) trophic group abundance and community diversity and (ii) SFW structure and function.

Abundance of HV, BV, FV, PR, and OV in each sample was counted and expressed per 100 cm$^3$ soil. Community diversity was analyzed using the Shannon diversity index [H$'$ = $-\Sigma Pi(ln\ Pi)$], where $Pi$ is the proportion of taxa in the total population [49], and used to compare diversity of either genera or trophic groups within the same community. Nematode trophic diversity based on the abundance of individuals in each trophic group was calculated using Hill's N1 effective number of abundant species formula [N1 = $exp(-\Sigma Pi\ (lnP_i))$], where $P_i$ is the proportion of trophic group $i$ in the total nematode community.

SFW structure and function of the soil groups were calculated as functions of EI = (100[e/(e + b)]) and SI = (100[s/(s + b)]) indices based on the weighted abundance of nematode guilds representing structure (s = $\Sigma K_s n_s$), enrichment (e = $\Sigma K_e n_e$) and basal (b = $\Sigma K_b n_b$) where $K$ is the specific weight of each guild and $n$ is the abundance of nematodes in each functional guild in the sample [28]. In simple terms, EI reflects changes in nematode population density relative to their food resource and reproduction and SI changes relative to resistance to disturbance. The graphical illustration of EI (*y*-axis) and SI (*x*-axis) to describe the SFW conditions in Quadrants A (enriched but unstructured), B (enriched and structured), C (resource-limited and structured), and D (resource-depleted with minimal structure) highlights the role of the c-p groups in a couple of ways. High EI values indicate that the soil is dominated by nematodes that are fast reproducing and resistant to disturbance (usually c-p 1 and c-p 2); whereas, high SI values indicate that the soil is dominated by slow reproducing and sensitive to disturbance (usually c-p 4 and c-p 5). Where both EI and SI are low (Quadrant D) indicates that the soil is degraded and depleted.

### 2.5. Data and Statistical Analyses

Four types of statistical analyses were performed. First, a generalized linear mixed effect model of PROC GLIMMIX [50] test of abundance of HV, BV, FV, PR, and OV and their sum, and diversity indices (H and N1) and EI and SI indices across countries (Ghana, Kenya and Malawi), soil group groups (Ferralsol, Lithosol and Nitosol), regions (north and south) and landscapes (undisturbed and disturbed) showed significant differences ($p < 0.05$) by country. EI and SI and diversity indices met test assumptions of homogeneity and normality, but abundance parameters did not and they were log-transformed (log x+1). The abundance and diversity parameters in soil groups, regions and landscapes were analyzed by country using the same generalized mixed effect model. The mixed model statistical inference was estimated using the containment denominator degree of freedom approach. Fields were always considered as the random effects and nested in fixed variables (country, region, soil group, and landscape). Least square means and standard errors of the abundance and diversity parameters by soil group, region and landscape (main effects) and the F-values of the two-way and three-way interactions are presented ($p < 0.05$).

Second, EI and SI between landscapes (undisturbed or disturbed) in each country (Ghana, Kenya and Malawi) by soil group (Ferralsol, Lithosol or Nitosol) and across soil groups was analyzed with PROC-GLIMMIX as described above. The least square means and standard errors ($p < 0.05$) were presented in a table format. The same least square means and standard errors ($p < 0.05$) of EI and SI of each country by landscape and soil group were fitted into the Ferris et al. model [28]. The EI and SI intersection point represents the location of the mean within a quadrant [28]. Overlap of the EI and SI standard error bars of the data points will show no significant difference within and across soil groups on the graph.

Third, as part of understanding the underlaying biological conditions associated with the soil health outcomes, the bivariate correlation among EI, SI, soil pH, SOC, percent sand, silt, and clay in undisturbed and disturbed landscapes by soil group and country were tested using Pearson correlation in PAST v4.03 software [51]. The correlation coefficient was calculated using:

$$r_{xy} = S_{xy}/S_x S_y \tag{1}$$

where $S_x$ and $S_y$ were the sample standard deviation and $S_{xy}$ the sample covariance. The coefficient was significantly different at $p < 0.05$.

Fourth, a principal component analysis (PCA) PAST v4.03 software was used to assess the composite relationships among HV, BV, FV, PR, and OV, and their sum, H and N1 indices, and EI and SI indices, and SOC, pH, and percent sand, silt, and clay across landscapes (undisturbed and disturbed), soil groups (Ferralsol, Lithosol and Nitosol), and countries (Ghana, Kenya and Malawi) [51]. Before performing PCA analysis with variance-covariance matrix [49], the mean data were standardized using:

$$z = (x - \mu/\sigma) \tag{2}$$

where $z$ was the standardized score, $x$ was the observed value, $\mu$ the mean of biotic or physiochemical parameter, and $\sigma$ the standard deviation. The outcome provides the best linear combinations of the variables along the horizontal axis (Principal Component 1, PC1) and vertical axis (PC2). A combination of PC1 and PC2 explains the total variability. Depending on how the different variables cluster or disperse provides information on how closely related the different soil groups may be.

## 3. Results

### 3.1. Nematode Abundance and Diversity

A total of 31HV,17BV,9FV,8PR, and 19OV taxa were detected. HVs were dominated by c-p 3, BV and FV by c-p 2, and PR and OV by c-p 4 (Table 1). Trophic group abundance varied by country, soil group, and landscape the most and within regions the least (Table 2). In all cases, herbivores were the most and predators the least abundant.

**Table 1.** A list of herbivore, bacterivore, fungivore, predator, and omnivore nematode genera and families and their colonizer-persister (c-p) values found in Ferralsols, Lithosols and Nitosols soil groups in Ghana, Kenya and Malawi [x,y].

| Trophic Group | c-p | Trophic Group | c-p | Trophic Group | c-p |
|---|---|---|---|---|---|
| **Herbivores** | | **Bacterivores** | | **Predators** | |
| *Cephalenchus* | 2 | *Panagrolaimus* | 1 | *Tripyla* | 3 |
| *Ecphyadophora* | 2 | *Rhabditis* | 1 | *Clarkus* | 4 |
| *Gracilacus* | 2 | *Acrobeles* | 2 | Mononchidae | 4 |
| *Paratylenchus* | 2 | *Acrobeloides* | 2 | *Mononchus* | 4 |
| *Psilenchus* | 2 | *Cephalobus* | 2 | *Seinura* | 4 |
| *Tylenchus* | 2 | *Cervidellus* | 2 | *Discolaimoides* | 5 |
| *Amplimerlinius* | 3 | *Chiloplacus* | 2 | *Discolaimus* | 5 |
| *Antarctylus* | 3 | *Eucephalobus* | 2 | *Nygolaimus* | 5 |
| *Criconema* | 3 | *Heterocephalobus* | 2 | | |
| *Criconemella* | 3 | Leptolaimidae | 2 | **Omnivores** | |
| *Criconemoides* | 3 | Monhysteridae | 2 | *Campydora* | 4 |
| *Discocriconemella* | 3 | *Plectus* | 2 | *Dorylaimellus* | 4 |
| *Helicotylenchus* | 3 | *Wilsonema* | 2 | Dorylaimidae | 4 |
| *Hemicycliophora* | 3 | *Zeldia* | 2 | *Dorylaimoides* | 4 |
| *Heterodera* | 3 | *Achromadora* | 3 | *Dorylaimus* | 4 |
| *Hirschmanniella* | 3 | *Prismatolaimus* | 3 | *Ecumericus* | 4 |
| *Hoplolaimus* | 3 | *Alaimus* | 4 | *Enchodelus* | 4 |
| *Meloidogyne* | 3 | | | *Labronema* | 4 |
| *Merlinius* | 3 | | | *Mesodorylaimus* | 4 |
| *Nagelus* | 3 | **Fungivores** | | *Paraxonchium* | 4 |
| *Paratrophurus* | 3 | *Aphelenchoides* | 2 | *Pungentus* | 4 |
| *Pratylenchus* | 3 | *Aphelenchus* | 2 | Actinolaimidae | 5 |
| *Rotylenchulus* | 3 | *Deladenus* | 2 | *Aparcelaimus* | 5 |
| *Rotylenchus* | 3 | *Ditylenchus* | 2 | *Aporcelaimellus* | 5 |
| *Trichotylenchus* | 3 | *Filenchus* | 2 | *Belondira* | 5 |
| *Trophurus* | 3 | *Diphterophora* | 3 | *Belondirella* | 5 |
| *Tylenchorhynchus* | 3 | *Leptonchus* | 4 | *Egtitus* | 5 |
| *Paratrichodorus* | 4 | *Tylencholaimus* | 4 | *Fuschelia* | 5 |
| *Trichodorus* | 4 | *Tyleptus* | 4 | *Prodorylaimus* | 5 |
| *Longidorus* | 5 | | | | |
| *Xiphinema* | 5 | | | | |

[x] A total of 152, 192 and 168 soil samples were analyzed from Ghana, Kenya and Malawi, respectively.
[y] Nematodes were classified [45] and assigned trophic groups [46,48] and colonizer-persister (c-p) values of 1 (colonizer) to 5 (persister) [26,28,47].

**Table 2.** Least square means and standard errors ($\pm$) of abundance of herbivore (HV), bacterivore (BV), fungivore (FV), predator (PR), and omnivore (OV) nematode trophic groups and their sum per 100 cm$^3$ of soil and diversity (Shannon, H, and Hill's, N1) indices observed in Ferralsol (FL), Lithosol (LL) and Nitosol (NL) soil groups (SG), across north (N) and south (S) regions (RG) and undisturbed (U) and disturbed (D) landscapes (LS) and interactions (F values) in Ghana, Kenya and Malawi.

| | Factors | | Nematodes Trophic Group | | | | | SUM | Diversity | |
|---|---|---|---|---|---|---|---|---|---|---|
| | | | HV | BV | FV | PR | OV | | H | N1 |
| Ghana | SG | FL | 28 ± 0.25 | 5 ± 0.14 a | 1 ± 0.19 | 1 ± 0.11 | 7 ± 0.14 a | 47 ± 0.17 | 1.6 ± 0.12 | 5.9 ± 0.75 |
| | | LL | 17 ± 0.23 | 4 ± 0.13 ab | 2 ± 0.18 | 0 ± 0.10 | 3 ± 0.13 b | 31 ± 0.16 | 1.5 ± 0.11 | 5.2 ± 0.71 |
| | | NL | 26 ± 0.20 | 3 ± 0.11 b | 2 ± 0.16 | 0 ± 0.09 | 4 ± 0.11 b | 42 ± 0.14 | 1.5 ± 0.10 | 4.9 ± 0.60 |
| | RG | N | 25 ± 0.20 | 5 ± 0.12 a | 1 ± 0.16 | 0 ± 0.09 | 5 ± 0.12 | 41 ± 0.14 | 2.0 ± 0.10 a | 5.9 ± 0.60 |
| | | S | 22 ± 0.16 | 3 ± 0.08 b | 2 ± 0.13 | 1 ± 0.07 | 4 ± 0.08 | 38 ± 0.12 | 1.0 ± 0.08 b | 4.8 ± 0.47 |
| | LS | U | 32 ± 0.22 a | 5 ± 0.12 a | 1 ± 0.17 | 0 ± 0.10 | 5 ± 0.12 a | 50 ± 0.15 a | 1.58 ± 0.11 | 5.5 ± 0.65 |
| | | D | 17 ± 0.14 b | 3 ± 0.08 b | 2 ± 0.11 | 0 ± 0.06 | 3 ± 0.08 b | 31 ± 0.10 b | 1.55 ± 0.07 | 5.2 ± 0.40 |
| | F values | RG*LS | 0.00 | 0.47 | 3.55 | 0.01 | 1.07 | 0.01 | 0.14 | 0.13 |
| | | SG*LS | 0.10 | 2.87 | 0.07 | 1.07 | 1.68 | 0.23 | 0.35 | 0.37 |
| | | RG*SG | 4.01 | 6.93 * | 1.21 | 0.59 | 9.42 * | 4.81 * | 0.02 | 0.49 |
| | | RG*SG*LS | 0.90 | 0.82 | 0.16 | 0.38 | 0.85 | 0.64 | 0.60 | 0.47 |

**Table 2.** *Cont.*

| Factors | | | Nematodes Trophic Group | | | | | SUM | Diversity | |
|---|---|---|---|---|---|---|---|---|---|---|
| | | | HV | BV | FV | PR | OV | | H | N1 |
| Kenya | SG | FL | 13 ± 0.18 b | 7 ± 0.15b a | 3 ± 0.16 ba | 0 ± 0.11 b | 3 ± 0.13 | 29 ± 0.16 b | 1.8 ± 0.08 | 7 ± 0.46 b |
| | | LL | 16 ± 0.18 b | 11 ± 0.15 a | 2 ± 0.16 b | 1 ± 0.11 a | 2 ± 0.13 | 37 ± 0.16 ba | 1.7 ± 0.08 | 7 ± 0.46 b |
| | | NL | 27 ± 0.18 a | 6 ± 0.15 b | 5 ± 0.16 a | 1 ± 0.11 a | 3 ± 0.13 | 47 ± 0.16 a | 1.9 ± 0.08 | 8 ± 0.46 a |
| | RG | N | 14 ± 0.15 | 5 ± 0.12 b | 3 ± 0.13 | 1 ± 0.09 | 2 ± 0.10 | 30 ± 0.13 b | 1.9 ± 0.06 | 7 ± 0.36 |
| | | S | 21 ± 0.15 | 13 ± 0.12 a | 3 ± 0.13 | 1 ± 0.09 | 2 ± 0.10 | 45 ± 0.13 a | 1.9 ± 0.06 | 7 ± 0.36 |
| | LS | U | 29 ± 0.18 a | 13 ± 0.15 a | 4 ± 0.16 a | 1 ± 0.11 | 3 ± 0.13 | 56 ± 0.16 a | 2a ± 0.08 | 8 ± 0.46 a |
| | | D | 11 ± 0.10 b | 5 ± 0.08 b | 2 ± 0.09 b | 1 ± 0.06 | 2 ± 0.07 | 24 ± 0.09 b | 1b ± 0.04 | 7 ± 0.25 b |
| | F values | RG*LS | 0.39 | 1.78 | 6.51 * | 1.60 | 3.67 | 0.73 | 0.51 | 1.86 |
| | | SG*LS | 0.67 | 0.72 | 1.98 | 0.43 | 3.64 | 0.91 | 1.32 | 2.56 |
| | | RG*SG | 1.55 | 0.94 | 3.22 | 1.50 | 4.09 * | 0.87 | 4.37 * | 4.08 * |
| | | RG*SG*LS | 0.23 | 3.18 | 1.07 | 0.61 | 3.24 | 0.29 | 3.21 | 2.83 |
| Malawi | SG | FL | 19 ± 0.37 b | 9 ± 0.15 b | 6 ± 0.19 b | 2 ± 0.11 | 5 ± 0.16 b | 51 ± 0.25 b | 1.6 ± 0.10 | 7.7 ± 0.62 |
| | | LL | 18 ± 0.38 b | 21 ± 0.16 a | 11 ± 0.20 ba | 2 ± 0.11 | 4 ± 0.16 b | 70 ± 0.26 b | 1.9 ± 0.11 | 7.1 ± 0.65 |
| | | NL | 92 ± 0.41 a | 16 ± 0.19 a | 16 ± 0.23 a | 2 ± 0.13 | 9 ± 0.18 a | 156 ± 0.28 a | 1.8 ± 0.12 | 6.4 ± 0.79 |
| | RG | N | 25 ± 0.30 | 13 ± 0.13 | 11 ± 0.16 | 2 ± 0.10 | 4 ± 0.13 b | 62 ± 0.21 | 1.74 ± 0.09 | 7 ± 0.51 |
| | | S | 40 ± 0.31 | 16 ± 0.14 | 10 ± 0.17 | 2 ± 0.10 | 7 ± 0.14 a | 95 ± 0.21 | 1.8 ± 0.09 | 7 ± 0.55 |
| | LS | U | 38 ± 0.39 | 13 ± 0.17 | 8 ± 0.21 b | 2 ± 0.12 | 7 ± 0.17 a | 75 ± 0.26 | 1.8 ± 0.11 | 7.5 ± 0.69 |
| | | D | 26 ± 0.21 | 16 ± 0.09 | 12 ± 0.12 a | 2 ± 0.07 | 4 ± 0.09 b | 78 ± 0.15 | 1.7 ± 0.06 | 6.6 ± 0.35 |
| | F values | RG*LS | 0.40 | 0.05 | 0.00 | 2.22 | 2.19 | 0.06 | 0.02 | 0.01 |
| | | SG*LS | 2.70 | 5.39 * | 0.32 | 3.82 | 0.29 | 0.19 | 0.34 | 0.58 |
| | | RG*SG | 0.15 | 7.98 * | 0.10 | 0.14 | 4.61 * | 1.11 | 2.65 | 1.62 |
| | | RG*SG*LS | 1.10 | 1.48 | 0.18 | 0.69 | 2.26 | 1.05 | 1.32 | 2.30 |

Nematode trophic groups were classified according to Yeates et al. [48] and Okada and Kadota [46]. Least square means with different letters within a column and factor category are statistically different at $p < 0.05$ among soil groups, between regions and between landscapes in each country. Data from mixed model based on containment degree of freedom approximation. Means with no letters are not significantly different. * F values significant at $p < 0.05$.

In Ghana, more bacterivores and omnivores were recovered in Ferralsols than in Lithosols and Nitosols. More herbivore, bacterivore and omnivore nematodes were recovered in undisturbed than in disturbed landscapes. Abundance of bacterivores and omnivores significantly varied by region and soil group.

In Kenya, Nitosols had more herbivores, fungivores and predators than either Lithosols and or Ferralsols; whereas, the reverse was true of bacterivores (Table 2). Only bacterivores varied by region. Abundance of herbivores, bacterivores and fungivores was higher in undisturbed than in disturbed landscapes. Interaction of region and landscape, and region and soil groups significantly influenced abundance of fungivores and omnivores, respectively.

In Malawi, herbivores and omnivores were most abundant in Nitosols than either Lithosols or Ferralsols; whereas, bacterivores and fungivores in Ferralsols were lower than those in Nitosols and Lithosols (Table 2). Abundance of bacterivores, by soil group and landscape, and bacterivores and omnivores, by soil group and region, were influenced significantly.

Shannon's diversity index (H) and Hill's effective number of abundant species (N1) were similar across soil groups in Ghana and Malawi (Table 2). In Kenya, effective number of abundant species was higher in Nitosols than in Lithosols and Ferralsols, and both effective number of abundant species and diversity were lower in disturbed than in undisturbed landscapes. Soil group and region affected both diversity and effective number of abundant species.

### 3.2. SFW Structure

The SFW data are presented in Table 3 and Figure 1. Table 3 presents SI (shade column) and EI (clear column) by country and landscape for each soil group. Within Ferralsols, SI was similar between landscapes in Ghana and Kenya, but significantly lower ($p < 0.05$) in disturbed than in undisturbed landscapes in Malawi. Within Lithosols, SI was significantly lower ($p < 0.05$) in disturbed than in undisturbed landscapes of Malawi. Within Nitosols, SI was significantly higher ($p < 0.05$) in undisturbed than in disturbed landscapes of Ghana and Malawi whereas, the reverse was true in Kenya. EI was similar between landscapes in all three countries and soil groups.

**Table 3.** Least square means and standard errors (±) of structure (SI, shaded) and enrichment (EI, clear) indices of undisturbed and disturbed landscapes of Ghana (GA), Kenya (KY) and Malawi (ML) in Ferralsol (FL), Lithosol (LL), and Nitosol (NL) soil groups (SG).

| SG | Country | Undisturbed | | Disturbed | |
|---|---|---|---|---|---|
| | | **SI** | **EI** | **SI** | **EI** |
| FL | GA | 80.3 ± 9.1 [ABC] | 13.7 ± 5.5 | 77.8 ± 6.6 [AB] | 18.5 ± 4.1 [ED] |
| | KY | 67.7 ± 8.4 [BCD] | 26.1 ± 4.8 | 68.2 ± 4.9 [BC] | 30.6 ± 2.8 [ABC] |
| | ML | 57.8 ± 8.4 a [D] | 35.7 ± 4.8 | 31.9 ± 4.9 b [E] | 36.0 ± 2.8 [AB] |
| LL | GA | 81.4 ± 9.1 [AB] | 26.0 ± 5.5 | 88.3 ± 5.8 [A] | 28.3 ± 3.4 [ABC] |
| | KY | 52.0 ± 8.4 b [ED] | 20.8 ± 4.8 | 62.4 ± 4.9 a [CD] | 26.0 ± 2.8 [CD] |
| | ML | 67.7 ± 8.7 a [BCD] | 29.3 ± 5.2 | 52.1 ± 5.0 b [D] | 37.0 ± 2.9 [A] |
| NL | GA | 85.8 ± 8.4 [A] | 32.0 ± 4.8 | 77.7 ± 4.9 [AB] | 27.7 ± 2.8 [BCD] |
| | KY | 38.5 ± 8.4 b [E] | 18.8 ± 4.8 | 62.8 ± 4.9 a [CD] | 10.8 ± 2.8 [E] |
| | ML | 54.1 ± 9.5 a [CD] | 19.2 ± 5.9 | 42.1 ± 5.4 b [E] | 31.3 ± 3.3 [ABC] |

The intersection of SI and EI within each landscape represents the data point for the country and soil group on Figure 1 [28]. Means followed by no or same letters are not statistically different at $p < 0.05$. Lowercase letters describe differences SI between undisturbed and disturbed and EI between undisturbed and disturbed landscapes for each country in a line. Uppercase letters describe differences among countries within and across soil groups in a column.

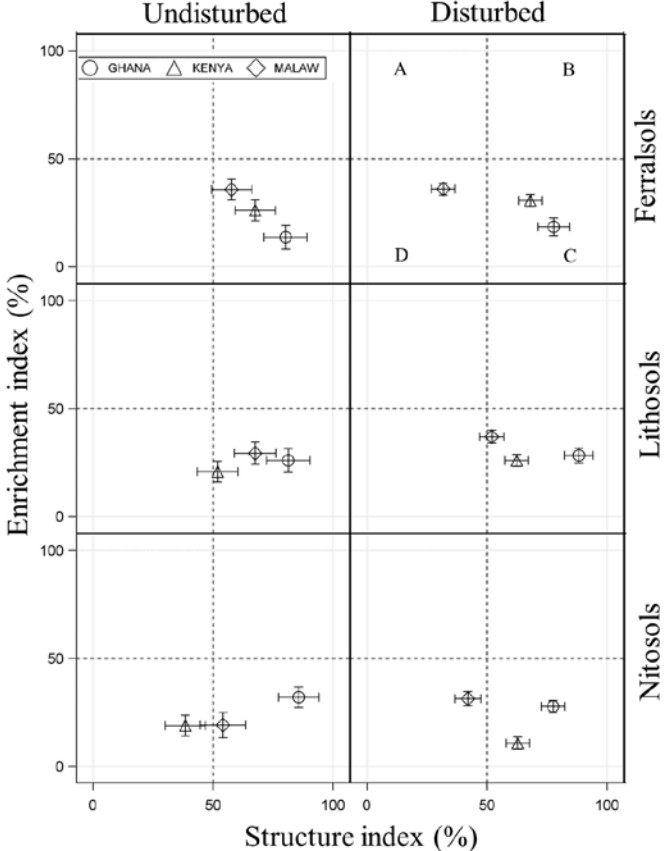

**Figure 1.** Soil food web structure of Ferralsols, Lithosols and Nitosols under undisturbed (left) and disturbed (right) fields of Ghana (circle), Kenya (triangle) and Malawi (square). The Quadrants A (enriched but unstructured), B (enriched and structured), C (resource-limited and structured), and D (resource-depleted with minimal structure) are based on the Ferris et al. [28] model. For purposes of simplicity, SFW structures of the undisturbed and disturbed landscapes are presented side-by-side.

Across countries within a soil group, SI was highest in undisturbed and disturbed Ferralsols in Ghana and lowest in Malawi ($p < 0.05$). EI was lower in the disturbed landscapes of Ghana compared with those of Kenya and Malawi ($p < 0.05$). EI was similar across countries in the undisturbed landscapes of Ferralsols. In Lithosols and Nitosols, SI was higher in Ghanaian than in Kenyan undisturbed landscapes, and higher in Ghanaian

than in Malawi disturbed landscapes ($p < 0.05$). EI was lower in the disturbed Nitosol landscapes of Kenya compared with those of Ghana and Malawi ($p < 0.05$). EI was similar across countries in the undisturbed landscapes of Lithosols and Nitosols.

Across countries and soil groups, SI of undisturbed Nitosols in Ghana was significantly higher ($p < 0.05$) than all but Ghanaian Ferralsols and Lithosols. In the disturbed landscapes, SI of Ghanaian Lithosols was significantly higher ($p < 0.05$) than all but Ghanaian Ferralsols and Nitosols. Kenyan Nitosols in undisturbed landscapes and Malawi Nitosols and Ferralsols in the disturbed landscapes had the lowest SI ($p < 0.05$). EI of the undisturbed landscapes was similar across the three countries and soil groups. EI of the disturbed landscapes was the highest in Malawi Lithosols and lowest in Kenya Nitosols (Table 3).

The EI and SI intersection point represents the location of the mean within a quadrant [28] where the alignments of the standard errors can be assessed visually. Overlapping of the data points of the graph depends on the significant differences of either SI and/or EI within and/or across landscapes and countries (Table 3). As there are overlapping data points, the undisturbed and disturbed landscapes are presented side-by-side (Figure 1).

In the undisturbed landscapes, the data points of Ferralsols in all three countries, Ghanaian and Kenyan Lithosols, and Ghanaian Nitosols fell in Quadrant C (resource-limited and structured), that of Kenyan Nitosols in Quadrant D (resource-depleted with minimal structure), and those of Malawi Nitsols and Kenya Lithosols borderline between Quadrants C and D (Figure 1). The Ghanaian Ferralsols data points with those of Malawi, Lithosols with those of Kenya, and Nitosols with those of Kenya and Malawi did not overlap.

In the disturbed landscapes, the data points for the Ghanaian and Kenyan Ferralsols, Lithosols, and Nitosols fell in Quadrant C, those of Malawi Ferralasols and Nitosols in Quadrant D, and that of Lithosols borderline between Quadrants C and D (Figure 1). The data points of the three countries and soil groups were not overlapping.

### 3.3. Bivariate Pearson Correlations of Biophysicochemical Parameters

Bivariate correlations of EI and SI indices, soil pH, SOC, sand, silt, and clay in undisturbed and disturbed fields in Ferralsol, Lithosol and Nitosol soil groups for Ghana, Kenya and Malawi are shown in Figures 2–4, respectively. Red circles indicate negative and blue circles positive correlations. The size of a circle is proportional to the degree of correlation. Circles in shaded squares show significant correlations ($p < 0.05$). The correlations varied by soil group and landscape within and across countries.

In Ghana, EI and SI were negatively correlated with each other only in undisturbed Ferralsols (Figure 2). In the undisturbed Lithosols, SI was negatively correlated with SOC and with silt (Figure 2). In the disturbed fields, EI was negatively correlated with SOC in Ferralsols and with sand in Nitosols, but positively correlated with clay in Nitosols. Correlation of SI was negative with pH and clay, but positive with sand in both landscapes of Lithosols.

In the undisturbed fields, soil pH and SOC were positively correlated with silt and clay in Lithosols, but they were negatively correlated with sand (Figure 2). In Nitosols, soil pH and SOC were positively correlated with each other (Figure 2). In the disturbed fields, soil pH was positively correlated with sand and silt in Ferralsols, SOC and clay in Lithosols, SOC and clay in Nitosols. Soil pH was negatively correlated with clay in Ferralsols, with sand in Lithosols and Nitosols (Figure 2).

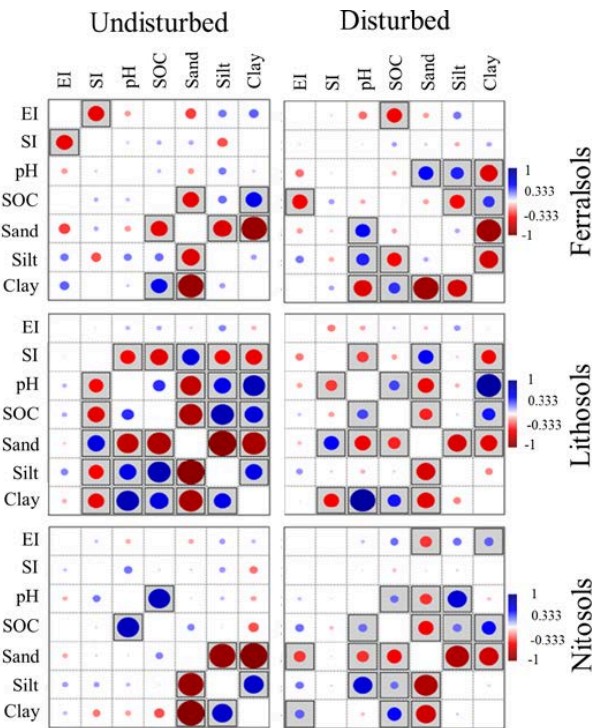

**Figure 2.** Bivariate Pearson correlations of EI and SI, soil pH, SOC, and percent sand, silt, and clay in undisturbed (left side) and disturbed (right side) fields of Ferralsol, Lithosol and Nitosol soil groups in Ghana. Red circles indicate negative and blue circles positive correlations. The size of a circle is proportional to the degree of correlation. Circles in shaded squares show significant correlations ($p < 0.05$).

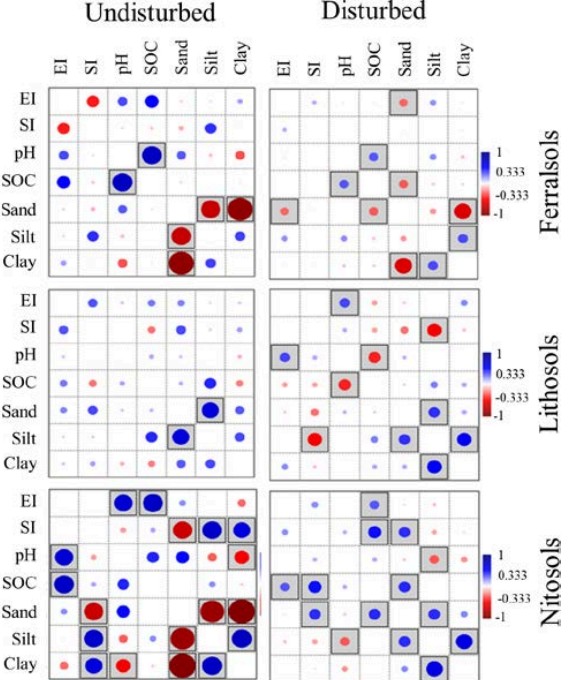

**Figure 3.** Bivariate Pearson correlations of EI and SI, soil pH, SOC, and percent sand, silt, and clay in undisturbed (left side) and disturbed (right side) fields of Ferralsol, Lithosol and Nitosol soil groups in Kenya. Red circles indicate negative and blue circles positive correlations. The size of a circle is proportional to the degree of correlation. Circles in shaded squares show significant correlations ($p < 0.05$).

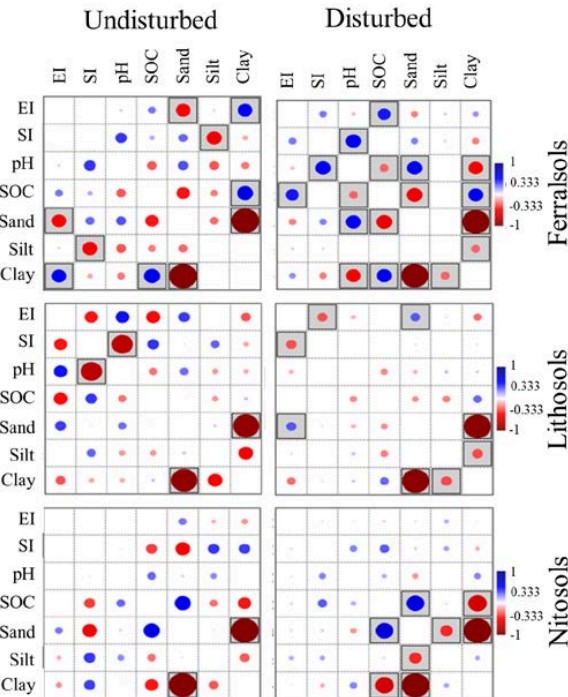

**Figure 4.** Bivariate Pearson correlations of EI and SI, soil pH, SOC, and percent sand, silt, and clay in undisturbed (left side) and disturbed (right side) fields of Ferralsol, Lithosol and Nitosol soil groups in Malawi. Red circles indicate negative and blue circles positive correlations. The size of a circle is proportional to the degree of correlation. Circles in shaded squares show significant correlations ($p < 0.05$).

In the undisturbed fields of Kenyan Nitosols, EI with pH and SOC and SI with silt and clay were positively correlated while SI was negatively correlated with sand (Figure 3). In the disturbed fields, EI was negatively correlated with sand in Ferralsols and positively correlated with soil pH in Lithosols and with SOC in Nitosols. SI was positively correlated with SOC and sand in Nitosols.

Soil pH and SOC were positively correlated with Ferralsols under undisturbed and disturbed fields while they were negatively correlated in Lithosols under disturbed fields (Figure 3). In Nitosol disturbed fields, soil pH was negatively correlated with silt and SOC positively correlated with sand.

EI in the natural vegetation Ferralsols of Malawi was correlated negatively with sand and positively with clay; SI was negatively correlated with silt in Ferralsols and with soil pH in Lithosols (Figure 4). In the disturbed fields, EI and SI were negatively correlated in Lithosols. EI was positively correlated with SOC in Ferralsols and with sand in Lithosols. SI was positively correlated with soil pH in Ferralsols (Figure 4).

In the natural vegetation, SOC and clay were positively correlated in Ferralsols while sand and clay were negatively correlated in all three soil groups (Figure 4). In the disturbed fields, soil pH was negatively correlated with SOC and clay and positively correlated with sand in Ferralsols. SOC was correlated positively with sand and negatively with clay in Nitosols (Figure 4).

### 3.4. Multivariate Principal Component Analysis

Multivariate biplot analyses of food web indices (EI and SI), soil physiochemistry (pH, SOC, sand, silt and clay), landscape (undisturbed and disturbed), soil group (Ferralsols, Lithosols and Nitosols), and country (Ghana, Kenya and Malawi) are shown in Figure 5. How the different variables align with principal components 1 and 2 (PC1 and PC2) provides a composite dimensionality of similarity (closer to one another) or dissimilarity (far apart) across the variables. The results show that the soil groups distinctly separated

by country, with those of Ghana along PC2, Kenya along PC1, and Malawi along PC1 but widely spread across PC2. Within Ghana, the three soil groups were distinctly separated, between landscapes, and only Nitosols were overlapping. In Kenya, the soil groups in the disturbed fields were more closely related than in the undisturbed fields. In Malawi, Nitosols in both landscapes and Lithosols in the disturbed fields were distinctly separated from the others. Sand, SI and SOC were aligned with the Ghanaian soil groups. The other variables were closely related to Malawi Ferralsols and Lithosols. A combination of PC1 and PC2 explains 61.4% of the total variability.

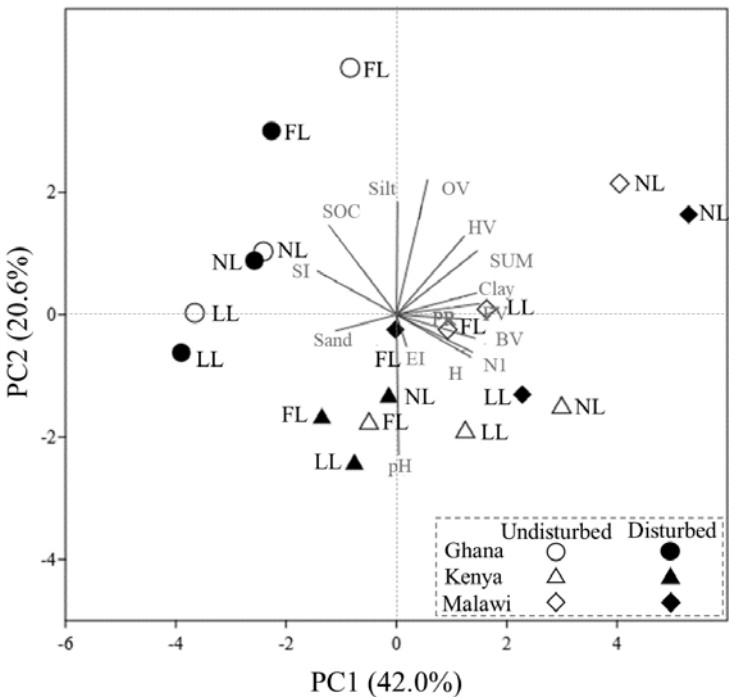

**Figure 5.** Principal component analysis of the relationships of nematode abundance (HV, BV, FV, PR, OV, and their SUM), diversity (H and N1), SFW (EI and SI), and soil physiochemistry (soil pH, SOC, and percent sand, silt and clay) of Ferralsol (FL), Lithosols (LL) and Nitosol (NL) soil groups in Ghana (circle), Kenya (triangle) and Malawi (square) under undisturbed (open signs) and disturbed (closed signs) landscapes.

## 4. Discussion

If soil health degradations in SSA or elsewhere can be managed with a one-size-fits-all or a location-specific approach is limited by variable standards and outcomes, lack of an integrated platform, and issues of scale within and across regions where the levels of soil degradations, cropping systems, land use practices, and cultures are highly variable [4,12]. This study established a first biophysicochemical proof-of-concept for location-specific approach when considering soil health management strategies in SSA. This study used a combination of nematode community, SFW structure, and soil properties and bivariate and multivariate analyses to characterize similarities and differences in soil health degradations of Ferralsol, Lithosol and Nitosol soil groups in Ghana, Kenya and Malawi. The nematode community and/or bivariate relationship analyses supported generally known variable results reports [10–14,33–36]. The SFW analysis separated soil conditions within and across categories of soil degradations and within and across soil groups and countries. The multivariate analysis of biotic and abiotic factors, soil groups and countries showed separations of the soil groups within and across countries.

### 4.1. Nematode Trophic Group Abundance and Diversity

The hypothesis that the soil groups have different nematode community abundance, composition and diversity is supported by varying abundance and community diversity and effective number of abundant species varying by soil group, region, landscape, and/or country. This is the first report on Ferralsol, Lithosol and Nitosol soil groups. Our variable nematode abundance and diversity data are generally similar to those reported in Arenesols, Cambisols and Vertisols in Kenya [33,34], Quartzite in Brazil [35], and Cambisol [36], Cambisol, Chernozem, and Stognosols [25], and Cambisol Fluvisols, Rigosols, Rendizina, and Stognosols [37] in the Slovak Republic.

The higher number of HV taxa than other trophic groups recovered across soil groups (Table 1) and the degraded and/or resource depleted SFW structure (Figure 1) support the notion that herbivore infestation and reproduction is severe in degraded soil conditions [48,52,53]. Among soil groups, total HV abundance appear to be generally higher in Malawi Nitosols than in Lithosols and Ferralsols in all three countries. This suggests that Nitosols may have more favorable conditions for nematodes than Ferralsols and Lithosols. At least in Ghana and in Kenya, the numbers of HVs and BVs were higher in undisturbed than in disturbed. Although unknown soil groups and variable experimental designs make generalized interpretation difficult, our data seem contradictory with those of Kimenju et al. [19] in Kenya and Talwana et al. [54] in Uganda who reported higher nematode abundance in disturbed than in undisturbed soils.

The low total numbers of BV, FV, PR, and OV trophic groups recovered in all soil groups in the three countries showed that the soils have common biological degradations. The population densities of the nematode trophic groups varying primarily by soil group and landscape suggests the stresses have different effects on the resident nematodes. What remains universally unknown is the exact population densities and combinations of nematode trophic groups needed for ideal soil health conditions for any soil and crop.

### 4.2. SFW Structure

It is generally known that high biodiversity is more desirable and a good indicator of soil quality [48,52–54]. A healthy SFW is supposed to consist of nematodes with different life strategies and feeding habits ranging from fast-growing, colonizer groups of nematodes at the bottom of the food chain, to slow-growing, persister groups at the top [55]. As stated in the previous section, there were similar numbers of taxa with differing trophic group abundance and overall low population density in the soil groups. What the SFW model does is to organize the taxa, population density, life history and sensitivity to disturbance information through the graphic relationship between EI and SI described in the four quadrants [28] and extract practical application out of complex biological data that otherwise would be difficult to understand.

The SFW analysis in this study revealed several major points. First, the hypothesis that the SFW structure of the soil groups are different is partially supported by where the data points fell in the four quadrants and the proximity of the data points within a quadrant. Most of the data points falling in Quadrants C, in both undisturbed and disturbed landscapes and all soil groups and countries, and in Quadrant D, in Malawi Ferralsol and Nitosol disturbed fields and Kenyan Nitosols in the undisturbed fields show that there are some differences among the soil groups. The separation of the data points within Quadrant C (resource-limited and structured) and/or Quadrant D (resource-depleted with minimal structure) by soil group and/or country, especially in the disturbed landscape suggests that the soils within the same quadrant may have different types and levels of stresses.

Second, the general expectation is that disturbed soils would be enriched [39,56]. However, the lack of data points in either Quadrant A or Quadrant B across all soil groups, landscapes, and countries indicates that none of the soil groups have enriched or agrobiologically suitable conditions. This is consistent with the well documented soil health degradations in SSA and elsewhere, albeit without corresponding nematode community analysis [10–12].

Third, the similarity in SFW conditions between the two landscapes suggests that the undisturbed soils may be as fragile as the disturbed soils. If this is due to the close proximity of the disturbed and the adjacent undisturbed sites or that the undisturbed soils are naturally fragile remains to be determined.

Fourth, there are few soil type or soil group-specific data base for SFW structure that can be used as the basis for developing soil health management practices. Moreover, it is not known what types of amendments will be needed or how long it will take to improve the soil conditions from where they are (data Quadrants C and D) to be ideal for agroecosystem (Quadrant B). The SFW structure data herein provide reference points that are helpful for achieving ideal agroecosystem conditions that generate desirable ecosystem services [39].

### 4.3. Bivariate Correlations

Understanding how indicators of the different soil health components relate to one another remains challenging because of disciplinary and cross-disciplinary integration gaps at the microenvironment and ecosystem level. For example, how changes in beneficial nematode composition (BV, FV, PR, and OV), soil microbiome, SOC, and soil structure as indicators of soil health [16,18,25,33,52] at the microenvironment level correlate with the ecosystem services that they generate [9–13,39] remains unknown.

The lack of and/or correlations without clear trend support the hypothesis that there is no consistent correlation of EI and SI with pH, SOC, texture (percent sand, silt and clay). The variable relationships among soil SOC, pH and sand, silt and clay observed herein reflect earlier reports. The EI and SI showed the least correlation with changes in clay, silt and sand texture soil pH, and most correlation with SOC. When correlations existed, there was no clear trend by landscape, soil group or country. In the disturbed fields for example, SOC in Ghanaian and Malawi Ferralsols was correlated with EI negatively and positively, respectively, while SOC was positively correlated with EI and SI in Kenyan Nitosols. In the natural vegetation, SOC was positively correlated with EI in Kenyan Nitosols, and negatively correlated with SI in Ghanaian Ferralsols and Lithosols.

The variable outcomes of our results generally confirmed the challenges of making generalized conclusions on bivariate correlations. Given that nematode abundance varied by soil groups, which, in turn, have non-overlapping SFW structures, however, the lack of consistent bivariate correlations of EI and SI is not surprising. Although EI and SI are indictors of process-based outcomes and provide valuable information, the relationship is not linear, and the relationship among the individual biophysicochemical indicators is highly variable. These messy variabilities suggested unaccounted confounding factors that cannot be explained by bivariate correlations alone.

### 4.4. Multivariate Correlations

Despite a substantial number of studies on the different components of soil health and ecosystem services from SSA and elsewhere [10,11,13,14,57], packaging the science in ways that can be applied in scalable ways remains challenging. The variable nematode abundance, SFW structure and bivariate correlations in this study, and meta analyses of ecosystem services in SSA and elsewhere further emphasize the challenges in understanding the fundamental biophysicochemical processes, much less scalable application. One way of overcoming the multitude of challenges is to build a database that integrates the biological, nutritional, physicochemical, structural, and water holding integrity components of soil health and test what patterns already existed and are yet to be scientifically generated.

This study integrated a portion of the belowground biological and physicochemical components of soil health and tested a hypothesis that there are similarities across the Ferralsol, Lithosol and Nitosol soil groups in Ghana, Kenya and Malawi through analyses of multifactor correlations. The separations of the parameters by country, soil group as well as landscape support the hypothesis that there are no similarities in the alignment of the measured parameters across the three soil groups and three countries. This suggests that the soil groups have different biophysicochemical properties, and raises an important

question. Since different soil groups may have overlapping soil textures (percent sand, silt and clay) [58] and few of the available soil health data consider soil groups, could not sorting data by soil groups be contributing to the variable results and confounding conclusions on soil health outcomes? Based on the data presented here, it is likely that more easily applicable soil health results may be obtained when quantifying parameters by soil groups than across soil groups.

The clustering of the different variables along PC1 and PC2 provided some insights of their relationships with soil groups within and across country. For example, SI, SOC and percent sand appear to separate along PC2 with the Ghanaian soils while the other biological parameters, silt and clay were closer to the Malawi soils. The separation of the Ghanaian soils along PC2, the Kenyan soils along PC1, and Malawi soils mostly away from PC2 shows that the same soil group in the three countries is different. The lack of overlapping, with the exception of Ghanaian Nitosols in disturbed and undisturbed landscapes, within a country and across landscapes shows that the soil groups have different biophysicochemical properties. The closer clustering of the Kenyan soil groups than those of Ghana and Malawi shows differences within countries. The close proximity of the Lithosols in Malawi and Kenyan disturbed landscapes possibly suggests similar impacts of land use practices.

Soil groups are the plates upon which different land use practices by different societies and cultures stamp their ecosystem footprints. Soil groups provide an opportunity for developing transformative and scalable models across climatic zones. To this end, this study establishes a foundation for building and/or developing scalable soil health knowledge, i.e., a combination of the alignment of the soil groups and biological and physicochemical parameters, along with the SFW data show that these soil groups have different biological structures and functions. The soil groups need to be treated separately rather than a one-size-fits approach. The study also demonstrates how useful the Ferris et al. [28] SFW model can be as a tool for identifying the status of the soil conditions and where they need to generate the desirable ecosystem services.

## 5. Conclusions

This study for the first time established a) varying number of taxa and abundance of nematode trophic groups by soil group, landscape and country, b) soil groups with mostly depleted and distinct SFW structure in both agricultural and natural landscapes across the countries, c) inconsistent correlation of EI and SI with soil physicochemistry, and d) distinct interaction patterns among nematode communities and soil physiochemical properties by country, soil groups and/or landscapes in Ferralsol, Lithosol and Nitosol soil groups in rural Ghana, Kenya and Malawi. The distinct patterns of interactions show that the soil groups have different biological properties and suggest that they may not respond the same way to a given treatment designed to ameliorate the existing soil degradations. Thus, this study established a proof-of-concept for a location-specific approach when developing management strategies that address the soil degradations in SSA.

**Author Contributions:** Conceptualization: H.M.; soil identification and processing: J.G., T.A.-G., G.N.K.; nematode identification: S.Y.: data analysis: I.L., Z.M.; writing—original draft preparation: H.M.; writing—review and editing: H.M., Z.M.; visualization: I.L.; supervision: J.Q., C.K., J.W.K.; project administration: H.M.; funding acquisition: H.M. All authors reviewed the manuscript. All authors have read and agreed to the published version of the manuscript.

**Funding:** This research was funded by The Howard G. Buffett Foundation; Grant No. RC101172.

**Institutional Review Board Statement:** Not applicable as this study did not involve humans or animals.

**Informed Consent Statement:** Not applicable.

**Data Availability Statement:** Data are not publicly available because they are part of an on-going study.

**Acknowledgments:** The authors thank an anonymous reviewer and H. Ferris, University of California, Davis, for constructive criticisms on the manuscript, V. Saka and M. Lowole and R. Mkandawire of Lilongwe University of Agriculture and Natural Resources, Lilongwe, Malawi for technical assistance, and T. Teal of MSU and A. Thuo of the University of Nairobi for assistance during sampling.

**Conflicts of Interest:** The authors declare no conflict of interest. The funders had no role in the design of the study; in the collection, analyses, or interpretation of data; in the writing of the manuscript, or in the decision to publish the results.

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
