# Peer review of "Nematode Community-Based Soil Food Web Analysis of Ferralsol, Lithosol and Nitosol Soil Groups in Ghana, Kenya and Malawi Reveals Distinct Soil Health Degradations"

_diversity, doi:10.3390/d13030101_

Round 1

Reviewer 1 Report

Thank you to the authors for sharing their manuscript. I understand and acknowledge that lots of time and hard work has gone into this document and I hope my comments will help improve it for publication. This manuscript is about understanding how soil food webs and other soil parameters across 3 soil orders in sub-Saharan Africa can characterize soil health with the goal to indentify management strategies for these soils. While I liked the combination of nematode food web indicators and soil variables (pH, OM, texture) for characterizing soil health, I do, however, think that the manuscript needs to be significantly revised for clarity and structure. One of the main issues is the lack of clarity around the nematode indices/SFW/c-p scaling. A nematologist well-versed in these methods might understand, but I think more ecological background and context for a more general audience is needed here to help the reader understand how and why this is a useful soil health metric. Additionally, the introduction will need additional information to provide support for the hypotheses – the reader should be able to understand why you are posing any hypothesis given some background in the text. I would suggest reorganizing the introduction to be more specific to your aims and adding specific background information that supports the hypotheses. Another main issue is with the statistics reported – as it is written, it is difficult to follow what statistical methods were used and what variables were included. I have provided specific details and comments below.

Abstract: I had difficulty understanding from the abstract what the main goal or takeaway of this manuscript is from the abstract. It sounds from the first few sentences like the main goal will be to develop management strategies for soil health, but as I keep reading, it looks like it might be to characterize the 3 of the 7 soil orders. I think it would be helpful to the reader to make some transition after the first two sentences to say why you need to characterize these soils first, and then at the end to tie it together back to soil health and management. I would clarify that you are doing 3 of the 7 orders, as the first sentence lead me to assume all 7 orders would be included. I’d also advise the authors to be careful in how they talk about EI and SI, which are specific nematode indices and could mean something very different if not defined clearly (e.g. outside of nematology, enrichment could mean fertilizer and structure could indicate aggregation/soil structure). The results reported in the abstract are vague and do not help the reader understand any main conclusion. These results also do not relate to the title – what are these distinct nematode food webs that the title mentions?

Specific comments on the abstract:

Line 24: We used the graphic relationship between enrichment (EI) and structure (SI) to characterize the soil food web (SFW)

-structure and enrichment mean very different things if you are a nematologist or a soil scientist. Is “Enrichment” and “Structure” that you mention in the abstract referring to a nematode index? Please clarify.

-is the soil food web specifically nematodes? please clarify.

Line 24: What is a “graphic relationship”? Is this a linear relationship? Please clarify.

Line 27: I don’t understand what is meant by an “integrated indicator”

Line 27-28: principal component analysis (PCA) to determine if the soil groups are similar to or different from one another”

- in what way (in their soil food web structure or soil factors, both)? Please report a result.

Line 31: I would suggest rewording this sentence for better clarity: “EI and SI correlated the most with %OM, but little consistency on trend, landscape, soil group or country.”

Introduction: The introduction will require reorganization. As it’s written, it’s hard to follow what the key points of the manuscript are about. It starts so broad that I initially think it’s about land management strategies for soil health relating to global problems. I think this opening paragraph could be edited to be more to the point. I think the main “problem” that this paper is trying to address is actually in the second paragraph “Despite a substantial knowledge on soil biophysiochemistry and agroecosystem science, however, there are few scalable models (add something here to clarify what these models are of…such as soil health/management??) that fit vast ecosystem degradations in SSA” with methods for achieving this appearing later (nematode indicators in paragraph 3 and limiting the study to 3 soil orders in paragraph 5). Perhaps the “three challenges” could be more succinct and moved up to the opening paragraph followed by a concluding sentence that tells the reader what this paper aims to do. I think these challenges are 1) current soil health management strategies have variable standards and outcomes, 2) soil knowledge that does exist lacks an integrated platform for soil health managers to use, and 3) developing management strategies is challenging due to issues of scale, variable land use across and within regions, and different levels of soil degradation. Please also make sure that you justify/provide reasonable evidence for your hypotheses (this can be in the introduction text, but the reader should read the hypotheses and understand why you hypothesized it based on what they just read). For example, there is a hypothesis that soil groups have the same nematode community after previous text states that soils are very different across scales – so why would nematode communities be the same? Finally, I would point out that soils were sampled from “subsistence agriculture and undisturbed conditions”, but that hypotheses do not mention this management treatment at all (later called these are called “landscapes”, please use consistent terminology throughout). I’d suggest adding expectations about how management/landscape affects soils to your hypotheses.

Specific comments on the introduction:

Line 41: “Soil health, defined as a given soil’s ability to function,”

-how does function in this sense relate to ecosystem function and services. It’s not clear throughout the manuscript where soil health and ecosystem function/services are being used interchangeably and when they are supposed to mean something different, such as the sentence below about organic matter (L. 47)

Line 45-46: “This, in turn, will require transformative and scalable soil health management strategies that fit the level and extent of degradations.”

  • Is that the purpose of this paper? To find scalable and transformative soil health management strategies?
  • Could you add an additional concluding sentence to this paragraph to suggest that important first steps would be to classify the soil food web/identify soil variables that link to foodweb/identify indicators in the soil food web or soil variables? I think this paragraph needs additional information to help the reader understand what the main goal of this manuscript will be.

Line 43-44: Please provide a reference for: “Unless the current soil health degradation in SSA is reversed or halted, providing the necessary ecosystem services to support all forms of life and long-term planetary sustainability will be difficult.

-Also, this is a huge statement –  as written, I take it to mean that our entire planet is in jeopardy if sub-Saharan African soil health continues to degrade. Please provide a reference for this

Line 47-48: Please provide a reference for: “An increase in percent soil organic matter (%OM) is one of the most common ecosystem service indicators in a given soil.”

– Also, why is it an increase? Should this say “soil organic matter content is a common ecosystem service indicator”?

-Secondly, as a topic sentence, this sets the expectation that the rest of this paragraph will be about organic matter, but I think it’s actually about soil health and management or land use. I’d suggest adding a new topic sentence to better transition into this paragraph and move this sentence about organic matter down to use as an example.

Line 48: “Despite a substantial knowledge on soil biophysiochemistry and agroecosystem science, however, there are few scalable models that fit vast ecosystem degradations in SSA [7-12].”

-Few scalable models of what? Organic matter content? Soil health? Ecosystem services?

Line 50: “There are at least three sets of challenges.”

  • Challenges to what? Building models of x? understanding soil health responses to management? Please clarify?

Line 52: “highly variable standards and outcomes” – will this manuscript be evaluating these methods?

Line 52-54: “This is not surprising given the heterogeneity of the land use practices across crops, soil groups, regions and socio-cultures, and that it is not yet possible to assess the footprints from a single core of soil”

-what is a ‘footprint’?

-I think this paragraph needs an additional sentence to wrap it up, it seems like it’s not fully developed as it’s written

Line 66-74: Please clarify the description of the SFW and the enrichment index and the structure index in terms that non-nematologists can understand. Readers may not be familiar with the quadrant visual and may find this description confusing.

  • Is the soil food web model a combination of these two indices?
  • How are these indices measured?
  • Do the SI and EI include some habitat measures of disturbance and food resources, respectively? (I know they do not, but it reads as if they do. I’d suggest clarifying this carefully to tell the reader about how they are interpreted - how different nematode groups are known to respond to certain conditions differently, so we can learn about the soil habitat by looking at the nematode community and without having to measure a bunch of soil factors).
  • How are the indices combined into the SFW model?

Line 75-82: This paragraph is underdeveloped.

Line 84: I don’t understand – what is a “ecosystem footprint stamp”?

Line 85: “The continent of Africa has seven major soil groups [31] that may serve as basis”

  • Are all 7 orders in SSA? Or across all Africa?
  • This setup leads me to believe that this study is about all seven orders. Why only 3 in the study? A better transition and explanation going from 7 to 3 is needed. Maybe a transition stating something along the lines of “Here, we focus on the 3 orders in SSA…”

Line 97: Why do you expect the soil groups to have similar nematode communities?

  • Previous paragraphs (e.g. lines 75-82) talked about how different soil is across scale, so why would you predict them to be the same?

Line 101: “If true, EI and/or SI along with the corresponding soil parameter will serve as integrated indicators of the soil conditions.”

  • How? I don’t understand this.

Line 102-104: “The fourth hypothesis was that there are similarities across the Ferralsol, Lithosol and Nitosol soil groups in Ghana, Kenya and Malawi.”

-Please be clear - similarities in what? Plants, nematode species, soil parameters?

Methods

The methods section needs significant revisions to help the reader understand what was done. In particular, I’d suggest adding explanations of the c-p scaling, EI, and SI in terms that non-nematologists can follow (providing meaning that will help a reader interpret any results in a ecological context). Statistics need to be clarified – what method was used (not which statistical package) along with the variables in each model. If mixed models were used, random effects need to be specified.

Specific comments on the methods:

Line 112-113: please use consistent terminology. Here you say “disturbed (agricultural and/or grazing)” and in the last paragraph of the introduction this was called “subsistence agriculture”

Lines 113-121: this text makes it difficult to follow the sampling setup/ how many samples were taken. The following paragraph (2.2. Sampling and Sample Processing) does help but this section still needs clarification so the reader can know what was done.

Line 124 – was each core processed separately or were these cores combined into one sample?

Line 128, 129, 131 – please clarify how many undisturbed/disturbed soil samples per soil group per region

Line 134: How do you take a 100cm3 subsample after sieving? This is inherently different that a 100 cm 3 sample from a core due to changes in bulk density. Did you weigh it? Please specify?

Line 137: did you measure percent soil organic carbon or soil organic matter? These are different things. If it’s percent organic carbon, I’d suggest changing your abbreviation to SOC? If organic matter, please clarify the text here.

Line 138: using standard procedures is fine, but it would be helpful to the reader if you tell us the name or a short description as there can be many different “standard” methods (organic carbon examples: 1) dry combustion after acidification, 2) Walkley-Black method, 3) loss on ignition, 4) soil respiration). I would separate out the references so they match up with each procedure.

Line 153: dry or wet soil?

Line 159-162: Please clarify this section for a more general reader, a non-nematologist will not understand how this is calculated or where these classifications come from. I would also add details about the c-p scaling so a reader can understand the first paragraph of the results section (e.g. what does it mean to have a cp value of 1 vs 5?). This would also be a good place to move the quadrant information from the discussion so the reader can interpret figure 1

Line 164: Please clarify the statistics here. PROC GLIMMIX is for generalized linear mixed models. Do you mean that you used generalied linear mixed models to test for differences using PROC GLIMMIX in SAS?

  • Also proc glimmix is for mixed models. Please clarify what the fixed and random effects were in this model?

Line 167-68: Some clarification is needed here to explain what is going on. Were model terms dropped/not presented here? What subsequent analyses? does this mean you performed some sort of model selection? I’m confused. It’s strange to drop terms from a model if they aren’t significant.

Line 169: change “where” to “were”

Line 184: principal component analysis is a multivariate analysis, you don’t have to specify multivariate here

-also, I’m confused about what was done here and I urge the authors to clarify their PCA approach carefully. As it currently reads, it seems that all variables were put into a PCA (both biotic and environmental together). A PCA is an unconstrained analysis, therefore you cannot correlate changes in environmental variables with biotic diversity. What you could do instead is a PCA to summarize the major patterns of variation within the environmental data only and then use canonical correspondence analysis (CCA) to examine the distribution patterns of nematode assemblages along the environmental and spatial gradients. CCA is a direct gradient analysis that uses both biotic and environmental data by combining ordination and regression techniques (see ter Braak 1986;Legendre and Legendre 1998).  

Line 188-191: I like how you helped the reader interpret PC results, something similar for the SFW analyses (c-p, EI, SI) above would be really helpful

Results

I found the results difficult to follow. I think this will need to be edited to clarify when the results are reporting an overall effect or when it is a within country effect. For example see sentence in line 198-199, “More herbivores…). Perhaps separate paragraphs about each country? Or even if it’s repetitive, put the country name in each sentence about that country? There are bits of methods throughout the results. Move these to the methods section, table/figure captions. Results sections should also report some statistics/effect sizes in the text (not simply “x and y were different”).

Line 200: delete “the” before “more”

Table 1. Please define J2 in the caption.

  • Are genera in italics and families not? Please clarify the taxonomy here.
  • Were all taxa found in all countries?

Table 2. Please add mean before abundance in the Table caption. Also, please define the abbreviations for nematode trophic groups and diversity indices in the caption.

  • Why do you not report a standard deviation or standard error?
  • F-values: only interactive effects? Did you not test for main effects of soil group, region, and landscape?
  • It would be helpful to report degrees of freedom for these models

Line 230: “describe the mean of the SFW structure” I don’t understand this. Mathematically, it’s not an average value, is it?

Line 229-232: “Description of the SFW has two components. First, the intersection of the enrichment (vertical) and structure (horizontal) indices describe the mean of the SFW structure of each of the natural and disturbed (agricultural) landscapes of the Ferralsol, Lithosol and Nitosol soil group in Ghana, Kenya and Malawi.”

  • This is methods.

Figure 1. It would be very helpful to the reader to define these quadrants at least in the caption or even printed on the figure. Something like: A) enriched, unstructured; B) enriched, structured, C) resource limited, structured, D) depleted, unstructured.

  • It is hard to see the difference between squares/diamonds and the circles

Line 242: Can you please clarify what you mean by separate? If this is just visually “separate”, I think that would be very subjective. Did you test if these were significantly different? If so, please clarify in the methods and here in the results.

Line 246: “The third…” I’m confused, the second sentence of this section said there are two components.

Line 255: do you mean “biophysicochemical”? please check spelling of this word throughout.

Figure 2A, B, C. I think naming these Figures 2, 3, and 4 would make more sense if they are presented separately in the manuscript.

  • Also these take up lots of space and present the same thing twice (each is a mirror image), is there a way to reduce redundancy here?
  • Please clarify in the caption that these are pearson correlations

Line 277, 282, Figure 2: correct spelling of ferralsol. Please check spelling throughout, this is where I caught the error, but I did not check thoroughly for spelling.

Line 311: spelling of “physiochemal”?

Figure 3. hard to read text on the figure. Perhaps pushing the labels out a bit from the lines and points would help?

Discussion

The discussion section largely reads like a results section (for example, see paragraph from 381-392 or 412-423 or 449-461) and in some cases like a methods section (for example, see lines 373-75, “The EI (y-axis) and SI (x-axis)…”). Please put the results in the context of your hypotheses and provide interpretation in the context of existing literature. Give details about what previous studies have found and tell us how your study is the same or different. What have you learned about these soils, and what could be done to improve soil health if that is your objective (lines 401-403)? In other words, tell us what these indices indicate should be done for management or what state the soil is, not just that the indices are useful for this type of thing. 

Line 347: what are the “known facts”, please help the reader here

Line 349-53: clarify what data are from this study and what are from others?

Line 357-358 – was the abundance of all trophic/c-p groups higher? Is this total abundance? Can you add some interpretation here?

Line 358 – what is a beneficial group?

Line 360 – what do you mean by stress?

Line 379: “well documented” and no citations?

Line 405: understanding?

Line 480-482: I don’t see how this conclusion makes sense. The previous results were all inconsistent across soils, etc.

Author Response

Please see the attachment below. 

Reviewer 2 Report

The study based on rich data sets from three Sub-Saharan countries is an attempt to provide a platform for the soil health diagnostics in order to be used for developing of appropriate management agricultural strategies.  To me the most important finding is that the authors showed that soil groups (types)/countries perform high specifity regarding nematode community parameters. This reflects the soil type genesis which is related to the geological history, climate and evolution of soil biota. Thus, one has always to take into account which soil groups are considered in order to use correctly the developed ecological indices based on terrestrial nematodes.

My criticism is related mainly to the literature source used to identify soil nematode taxa, which reflects at least one index - H. Using Andrassy’s books (Andrassy, 2005, 2007, 2009), would be more appropriate since they encompass world nematode fauna.  

No data about different parameters variability between separate samples since mean values are used.

I wonder if data on N content in soil have been collected and used in the analyses performed.

Mapping the studied parameters in geographical ranges studied is recommended to help visualization of the observed patterns.

Reference 55 is not in the reference list.

Some small notes are in the text.

Author Response

Please see the attachment below. 

Round 2

Reviewer 1 Report

Thank you to the authors for sharing this manuscript. While I think that this version is an improvement on the previous one, the manuscript will need additional revisions before it will be ready for publication. In the most polite way I can possibly say this - I am not pleased that many of my comments need to be repeated here from my previous review, as the authors did not address them and did not comment on why they did not address them (even very simple ones, such as correcting the spelling of one of the soil groups).

Abstract: The abstract has improved greatly, thank you to the authors for making it much clearer. I would still suggest more specific language in the results sentences, and I give one example below, but suggest each sentence be carefully addressed.

-biophysicochemical is spelled wrong throughout the manuscript, as was previously pointed out

Line 25: Again, I still don’t know what is meant by “graphic relationship”, please reword

Line 25: “We used a graphic relationship between changes in nematode population dynamics relative food 25 and reproduction (enrichment, EI)” should this say “… changes in nematode population dynamics relative to food source…”

Line 33: For results, can you be more specific? For example, this sentence“The resource-limited and degraded SFW conditions distinctly separated by soil groups” could be changed to “While the SFW of all three soil groups indicated resource-limited and degraded conditions, SFW measures distinctly separated by soil group within this classification. “

Introduction: The text of the introduction is easier to follow, thank you to the authors for making improvements. I have specific comments, most of which were not addressed from my previous review, which I list again in bold below. Most problematic here, I do not think the hypotheses logically follow this introduction and I ask the authors to please clarify their expectations for differences in soil groups in the introduction – why, specifically, would you expect three different soil orders to have the same nematode communities? If these soil orders all have the same pH and texture, that could be a justification, for example. As it’s currently written I would expect all soil orders to have different communities.

Copied from original review, as this was not addressed. “Function” is a vague term and needs definition

Line 45: “Soil health, defined as a given soil’s ability to function,”

-how does function in this sense relate to ecosystem function and services? It’s not clear throughout the manuscript where soil health and ecosystem function/services are being used interchangeably and when they are supposed to mean something different, such as the sentence below about organic matter (L. 47)

Line 55-61: paragraph needs a closing sentence to transition to the next

Again, I still think that no one except a nematologist will be familiar with or able to picture the SFW quadrants as written. I still think that this is more appropriate to the methods section.

Line 76-79: “Ferris et al. [28] used the relationship between changes 76 in population density of BV and FV nematodes relative to food resource and reproduction 77 (Enrichment Index, EI) and resistance to disturbance (Structure Index, SI) to describe the SFW 78 conditions in four quadrants. These are: enriched but unstructured (A), enriched and structured (B), 79 resource-limited and structured (C), or resource-depleted with minimal structure (D).”

-If this must be kept in the introduction, perhaps change to something like, “To visually describe the SFW, Ferris et al. [28] plotted the nematode Enrichment Index (EI; y-axis) , which evaluates changes in population density of nematodes by functional guilds to relative to their food resource and reproductive rate, against the Stucture Index (SI; x-axis), which measures changes in these nematode functional guild population densities relative to their resistance to disturbance. In this way, four SFW conditions are identified in x-y quadrants: enriched but unstructured (quadrant A; negative-x, positive-y), enriched and structured (quadrant B; positive-x,positive-y), resource-limited and structured (quadrant C, negative-x, negative-y), or resource-depleted with minimal structure (quadrant D; positive-x, negative-y).

Line 77: I missed this in the previous version, but to my knowledge the EI and SI do not only include bacterivores and fungivores, but also include omnivores and predators (or “carnivores” as Ferris et al 2001 calls them).

Repeating from my previous review, Why do you expect the soil groups to have similar nematode communities?

Line 106-108: “However, the limited information on Arenesols, Cambisols and Vertisols in Kenya shows that nematode abundance and diversity vary by soil groups [33]. Our first hypothesis is that the soil groups have similar nematode community abundance and diversity.”

-The first sentence is about how different nematode abundance/diversity is by soil group for 3 other soil groups in Kenya, so why here in the very next sentence would you predict these 3 groups to be the same?

Hypothesis 2 (and1, and  3, and 4): Why do you expect these 3 soil orders to be the same? Are they similar texture, pH, land use, plant communities, climate? Please add some justification/background on these soils for this in the introduction. My understanding is that different soil properties found amongst types/orders very often have different nematode communities, see example references:

  • Lišková, M. & Čerevková, A. & Hanel, Ladislav. (2008). Nematode communities of forest ecosystems in association with various soil orders. Russian Journal of Nematology. 16. 129-142.
  • Lima da Silva, J.V.C., Hirschfeld, M.N.C., Cares, J.E., Esteves, A.M. (2019). Land use, soil properties and climate variables influence the nematode communities in the Caatinga dry forest. Appl. Soil Ecol., 103474. DOI: 10.1016/j.apsoil.2019.103474
  • And, already referenced here, also contradicts your hypothesis: Thuo, A.K.; Karuku, G.N.; Kimenju, J.W.; Kariuku, G.M.; Wendot, P.K; Melakeberhan, H. Seasonal 620 variation of nematode assemblage and diversity on selected soil groups in Kenya: Vertisols, Cambisols and 621 Trop. Subtrop. Agroecosystems. 2020a, 23, #63.

Line 113-115: “A similar SFW structure would be where the data points for the soil groups fall in either Quadrant A, B, C or D.” I don’t understand how this would be a similar sfw structure. Do you mean that all the data points for all soil groups fall in the same quadrant? Can you please clarify?

Note the change in verb tense from hypothesis 1 to the other hypotheses

Methods:

Repeating from previous review: what does it mean to have a cp value of 1 vs 5?

Line 146-47, 151-68: Repeating from previous review, consistent terminology, please. “agricultural field” and “adjacent natural vegetation” – should be changed to “disturbed” and “undisturbed landscapes”, respectively. Please check this throughout the manuscript to avoid confusion.

Line 167: Again, How do you take a 100cm3 subsample after sieving? Do you have bulk density data? Or are you assuming that 100mL of soil = 100 cm3, as one would assume for water? We need a reference here for the method, please.

Line 223: instead of “graphically plotted” can you say something about how structure is on the x-axis and enrichment index is on the y-axis following Ferris et al 2001.

Line 234-41: Regarding the PCA - Repeating from my previous review: This section needs additional clarifying details to understand the PCA methods. One problem with combining all these data into one PCA is that they are all in different units and scales (especially environmental data), and data are often standardized prior to PCA. Another problem is that biotic data often have non-linear responses to environment (like linear regression, PCA assumes normality) – thus biotic data are often transformed for PCA if necessary.

-Were variables standardized or Transformed here?

- was the PCA performed on the variance-covariance matrix or the correlation matrix?

Again, It is not statistically appropriate to put both biotic and environmental variables together into one PCA.

Results: It is very hard to follow the results as they are written and there has been little improvement on this. Repeating from previous review: I think this will need to be edited to clarify when the results are reporting an overall effect or when it is a within country effect. For example see sentence in line 249, “More herbivores…). Perhaps separate paragraphs about each country? Or even if it’s repetitive, put the country name in each sentence about that country? Spelling corrections that I previously pointed out were not made. Please correct the spelling.

Line 285: Please revise this sentence for grammar: “In order to simplicity separation of the data points within a quadrant”

Line 296: delete “were distinctly separated”

Line 295-297: “Within the quadrants, a reference to the standard deviation bars revealed a distinct separation (dissimilarity) of the agricultural landscapes within soil groups of all three countries were distinctly separated (Figure 1).” – How can you possibly see that landscapes differ by country when they are on separate plots if your metric is those standard error bars? I would suggest a statistical test to provide support for this claim or revising the plots so the reader can see that they do differ

Table 2: A standard deviation or error is needed to give the reader an idea about variability, I don’t think it matters if the table is untidy or busy. It’s about being transparent about the variability of the data. Similarly, degrees of freedom are needed to be able to interpret F-statistics (most other journals require this).

  • And again, from my previous review: F-values: why only interactive effects reported? Did you not test for main effects of soil group, region, and landscape?

Figure 1: check spelling in the caption

Figure 2, 3, 4, Once again: Please clarify in the captions that these are pearson correlations

Discussion: I appreciate the authors more specifically addressing their hypotheses in this revision. I think this discussion still needs to provide clearer interpretation of the results and to put the results/interpretation in the context of existing literature. Here I give one example, but the authors will need to carefully revise the full discussion. For example, line 443, “In this study they were not” – why not? Please help the reader understand. Are fertilizers not used in subsistence agriculture, is the soil nutrient depleted, or very old soil perhaps? Give details about what previous studies have found and tell us how your study is the same or different. Or, if there are no previous studies, tell us what factors about the landscape/climate/soil could contribute to this. Help the reader interpret the data.

Author Response

See the file below 
